# Spontaneous and Controlled Macroscopic Chiral Symmetry Breaking by Means of Crystallization

**Gérard Coquerel \*** and **Marine Hoquante**

SMS EA3233, Place Emile Blondel, University of Rouen Normandy, CEDEX, F-76821 Mont-Saint-Aignan, France; marine.hoquante@univ-rouen.fr

\* Correspondence: gerard.coquerel@univ-rouen.fr

**Abstract:** In this paper, macroscopic chiral symmetry breaking refers to as the process in which a mixture of enantiomers departs from 50–50 symmetry to favor one chirality, resulting in either a scalemic mixture or a pure enantiomer. In this domain, crystallization offers various possibilities, from the classical Viedma ripening or Temperature Cycle-Induced Deracemization to the famous Kondepudi experiment and then to so-called Preferential Enrichment. These processes, together with some variants, will be depicted in terms of thermodynamic pathways, departure from equilibrium and operating conditions. Influential parameters on the final state will be reviewed as well as the impact of kinetics of the R ⇔ S equilibrium in solution on chiral symmetry breaking. How one can control the outcome of symmetry breaking is examined. Several open questions are detailed and different interpretations are discussed.

**Keywords:** chirality; deracemization; preferential enrichment; thermodynamics; phase diagrams; kinetics

## 1. Context, Introduction

### 1.1. Chiral Discrimination between Pairs of Enantiomers in the Solid State

Two behaviors of the R–S system regarding racemization need to be distinguished. In the first case, the two enantiomers do not interconvert under the operating condition or in the time scale of the experiment. Therefore, the system is a symmetrical binary system where the two components have exactly the same thermodynamic properties (temperature, enthalpy and entropy of fusion, density, Cp versus T, etc.). This behavior is represented in Figure 1A–D. In the second case, the two enantiomers racemize rapidly in the liquid state. There is a relationship of interdependence between the two components, so the system is actually a degenerated binary system, as depicted in Figure 1E.

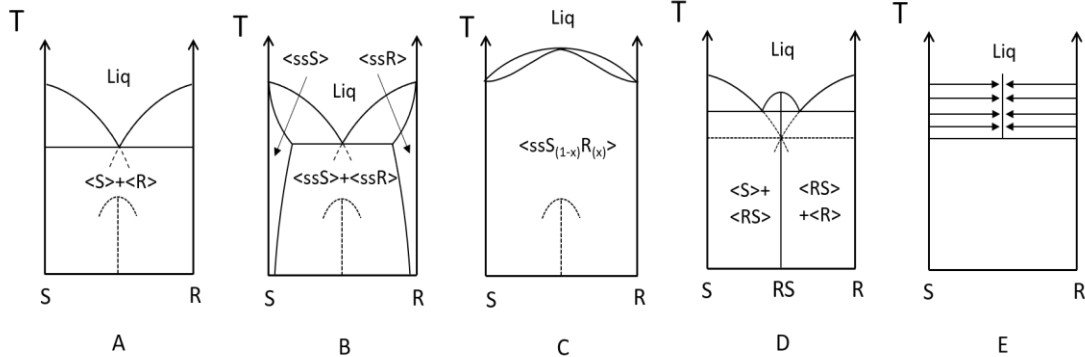

**Figure 1.** Binary system between a pair of enantiomers showing different types of chiral discriminations in the solid state: (**A**–**D**) in the absence of racemization in the liquid state; (**E**) with racemization in the liquid state. In cases (**A**–**C**), the vertical dashed lines represent metastable racemic compounds. System E represents a conglomerate with fast racemization in the liquid state.

There are several degrees of chiral discrimination in the solid state [1]. The top chiral discrimination occurs when every single crystal contains a single enantiomer, which is called conglomerate. This is often defined as "spontaneous resolution" (Figure 1A). The same case can exist with a partial solid solution (Figure 1B). By extension, there is also the possibility of a complete solid solution with a maximum or minimum point where, in the full composition range, both enantiomers are randomly distributed over the crystallographic sites. (Figure 1C). In the latter case, if there is no miscibility gap in the solid state, the chiral discrimination in this single solid phase is very poor. Then, there is the important heterochiral recognition corresponding to the formation of a stable <1-1> stoichiometric intermediate solid phase, called the "racemic compound" (Figure 1D). The latter case is by far the most popular: it accounts for 90–95% of all derivatives of a given couple of enantiomers. Crystallographic surveys show that most of these racemic compounds crystallize in a centro-symmetric space group ($P2_1/c$, P-1, Pbca, C2/c have the greatest occurrence). Nevertheless, there are several hundreds of kryptoracemic compounds (KRCs) in CSD version 2020. These KRCs account for ca. 1% of the racemic compounds. In those structures corresponding to chiral space groups (also named Sohncke space groups), the two enantiomers are in equal amount in the unit cell, but as independent molecules [2–5]. Therefore, Z' is an even number. When Z' > 2, other possibilities can arise such as anomalous conglomerates [6–8].

When changing the temperature, the chiral discrimination can be slightly or even completely altered. Indeed, from a racemic compound at low temperature, a stable conglomerate can be obtained at higher temperature through a three-phase peritectoid invariant [1]. The opposite situation is also well known. The three-phase invariant is then a eutectoid [9]. The switch from a racemic compound to a conglomerate-forming system can also appear with the addition of a particular solvent or co-crystal former or both [10]. The addition of crystal co-formers (isolated or in a mixture of solvents, counter-ions, co-crystal former, etc.) greatly increases the number of possibilities to explore in order to spot at least one conglomerate [11,12]. This increases the order of the system, which is no longer binary, but ternary, quaternary, quinary, etc. [13]. High-throughput techniques are of great help to alleviate the amount of work due to the overwhelming number of tests [14,15].

A complete solid solution at high temperature does not prevent the existence of a large miscibility gap in the solid state at low temperature [16]. The chiral discrimination in the solid state increases continuously as the two symmetrical solvus curves become more apart and towards a low temperature.

We will see that (paragraph on Preferential Enrichment: PE hereafter), for some special cases, and for systems initially very far from equilibrium reproducible chiral symmetry breaking can be observed. Conversely, departure from equilibrium can be detrimental to the chiral discrimination in the solid state. Fast cooling can lead to solid solutions without PE effects and even more severe

cooling (i.e., quenching) can lead to amorphous material without any kind of macroscopic chiral recognition [17].

Partial solid solutions between enantiomers are also well known [5,18]. Those cases are intermediate between conglomerates without solid solution and with complete solid solution. In some systems, the crystal growths of the pure enantiomers can lead to peculiar microstructures named lamellar conglomerates, which should not be confused with racemic compounds [19].

*1.2. Equilibrium in Solution*

In solution, we can consider two different extreme situations: no racemization and fast racemization [10]. In the former, the chirality of the chemical entity is blocked in the solid state as well as in the liquid state. The latter encompasses: (i) the loss of chirality in a solution such as sodium chlorate ("racemization" is instantaneous here); (ii) some atropisomers with a low energetic barrier between the enantiomers in solution; (iii) enantiomers that can rapidly interconvert by the action of a catalyst (a base, an acid, an enzyme, etc.) (Figure 2).

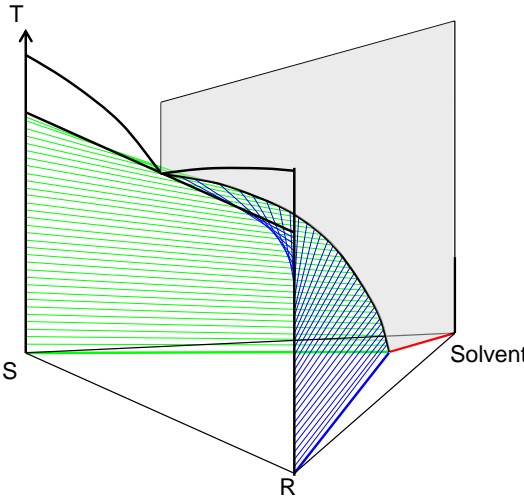

**Figure 2.** Degenerated conglomerate-forming ternary system with fast racemization in the liquid phase. The vertical plane contains the racemic liquor from saturated solution at a different temperature to the vertical line of the pure solvent. All other compositions of the liquid phase are simply not accessible. The green and blue tie lines, respectively, connect the S enantiomer and the R enantiomer to the saturated racemic solution at a given temperature.

## 2. Macroscopic Spontaneous Chiral Symmetry Breaking Induced by Crystallization

*Deracemization Induced by a Flux of Energy Crossing the Suspension (DIFECS)*

When a fast racemization takes place in solution under the same condition as the crystallization of a conglomerate, it is possible to observe a spontaneous macroscopic break in symmetry. The corresponding phase diagrams are no longer those displayed in Figure 1A–D, as detailed in [20,21]. Indeed, the phase diagrams are degenerated because the liquid phase can only contain an equimolar amount of enantiomers (i.e., a racemic composition see Figures 1E and 2). Under those circumstances, any energy flux passing through the suspension for long enough will lead to the disappearance of one population of homochiral crystals. A constant mechanical stress such as grinding (known as Viedma ripening) [22], numerous temperature cycles [23], long exposure to ultrasound [24], pressure stress [25] or microwaves [26] are general methods enabling the evolution of the initial dual population of particles to a single population of crystals containing a single enantiomer only, i.e., deracemization. Those methods operate rather close to thermodynamic equilibrium.

In Figure 3, the isotherm shows the main feature of the process. V stands for the solvent, while S and R are the two enantiomorphous chemical entities. Due to the fast racemization in solution or simply the absence of chirality in solution, the attainable states of the system are inside the triangle S-R- $L_{SAT}$ and along the racemic line V- $L_{SAT}$. $L_{SAT}$ is the point representative of the saturated racemic solution at that temperature. Under strong enough continuous attrition, the initial suspension represented by point I evolves towards $F_R$ or $F_S$ (Figure 3A,B). Simultaneously, the racemic mixture of solids represented by point M evolves towards S or R, i.e., the pure enantiomers. The tie lines connecting the constant saturated liquid $L_{SAT}$, the point representative of the overall synthetic mixture and the point representative of the solid composition move from $L_{SAT}$–I–M to $L_{SAT}$–$F_S$–S or $L_{SAT}$–$F_R$–R. If the system does not contain any chiral impurity and the initial two populations of crystals are symmetrical in terms of Crystal Size Distribution (CSD) and Growth Rate Dispersion (GRD), the last stage of evolution, that is, crystals of pure S or else pure R in equilibrium with $L_{SAT}$, is purely stochastic. Usually, the kinetics of this spontaneous evolution are of the first order. In other words, logarithm of enantiomeric excess (e.e.) versus time is linear. The more the overall composition departs from 50–50 composition, the faster the evolution towards homochirality is. Thus, it is an auto-catalytic process. However, if the system contains chiral impurities, the growth and dissolution rates of the two enantiomers become different, promoting one enantiomer over the other. The final evolution of Viedma ripening can be directed by using a chiral impurity and the e.e. versus time can also evolve linearly [27].

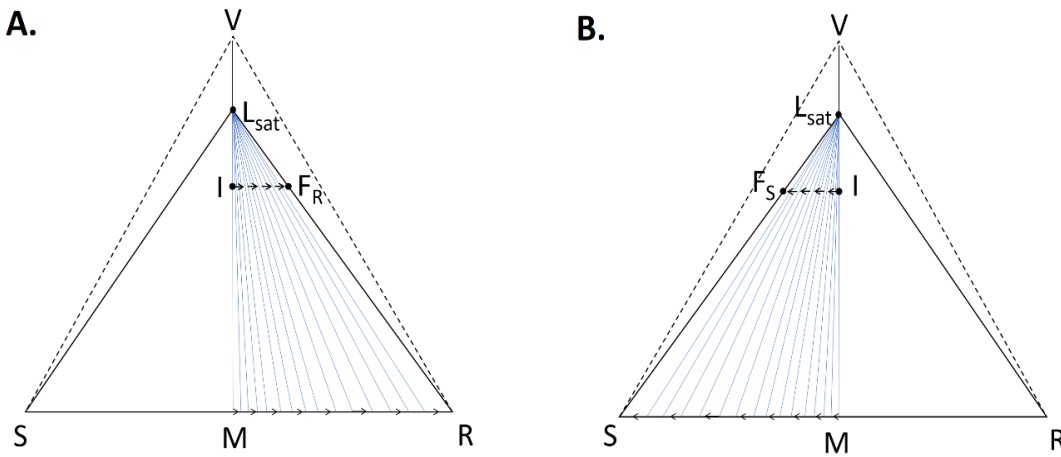

**Figure 3.** Spontaneous deracemization by Viedma ripening, ultrasound, microwaves towards R enantiomer (part (**A**)) and towards S enantiomer (part (**B**)). Starting from a suspension represented by point I, composed of a doubly saturated solution $L_{SAT}$ and an equimolar mixture of pure chiral solids, the system will spontaneously break its symmetry by an evolution towards $L_{SAT}$–$F_R$–R (represented) or conversely to a symmetrical system: $L_{SAT}$–$F_S$–S (not represented). $F_R$ and $F_S$ represent the two possible final compositions of the overall synthetic mixture after symmetry breaking.

Viedma ripening can be implemented directly during the synthesis; this method is called asymmetric synthesis, involving dynamic enantioselective crystallization [28].

For example, isoindolinones, a class of compounds used as core structures for pharmaceutical applications, were resolved successfully using Viedma ripening by the group of Vlieg [29]. Indeed, isoindolinones 1–3 (Figure 4) crystallize in a conglomerate-forming system and racemize quickly in solution without a catalyst.

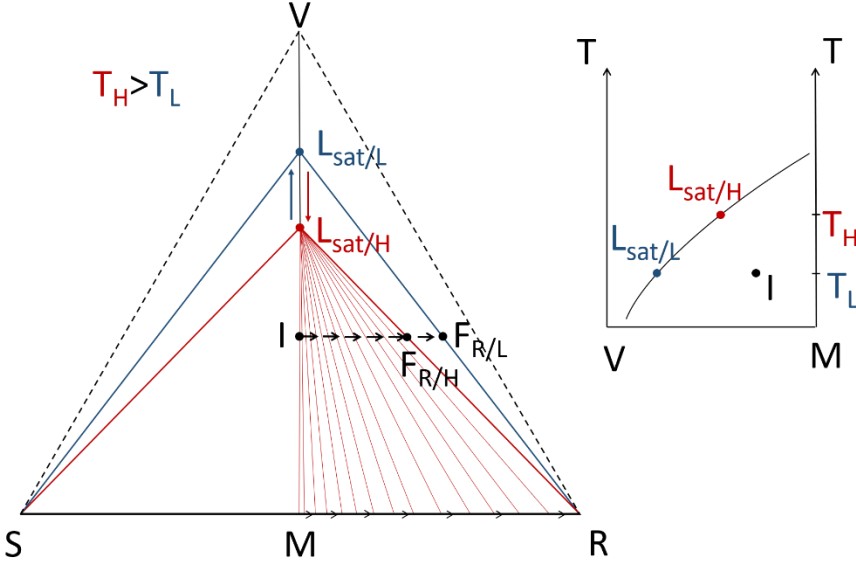

**Figure 4.** Isoindolinones that could be deracemized by Viedma ripening.

Repetitive oscillations of temperature are also able to induce deracemization in the system without limiting scale. This spontaneous process is known as Temperature Cycle-Induced Deracemization (known as TCID) [23]. The temperature gradient could be in space [30] or in time [31], or even both [32]. Figure 5 shows the corresponding phase diagram, which illustrates the evolution of the system when the temperature of the whole system fluctuates between $T_L$ and $T_H$, the low and high temperatures, respectively. When the e.e. of the solid phase strongly departs from zero, it is beneficial to decrease the amplitude of the temperature oscillation. The damped effect avoids wasting time and energy by preventing the dissolution of the enantiomer in excess [33].

**Figure 5.** TCID experiment: the inset shows the racemic vertical section versus temperature M–V–T, starting from a suspension of composition I, composed of a racemic mixture M of pure enantiomers in equilibrium with their saturated solution at $T_L$. Repetitive thermal oscillations of the suspension from $T_L$ to $T_H$ back and forth lead to complete deracemization (i.e., chiral symmetry breaking). Here, only the evolution to R is represented, but, in the absence of any chiral bias, the system can evolve to S as well. The amplitude and kinetics of the temperature oscillation impact the kinetics of deracemization.

Figure 5 schematizes the principle of deracemization by TCID; only one spontaneous evolution is shown: towards the R enantiomer. Two isotherms are represented: one at $T_L$ and the other one at $T_H$. When the initial suspension I is put back and forth from $T_L$ to $T_H$, it undergoes partial dissolution and recrystallization cycles. Starting from a suspension represented by point I, the overall synthetic mixture evolves towards $F_{R/H}$ at $T_H$ and $F_{R/L}$ at $T_L$. Simultaneously, the composition of the solid phase evolves towards the pure enantiomer R (or S; the latter case is symmetrical to the one represented in Figure 5). It is worth noting that there is an initial period without a noticeable evolution in the e.e. of the solid phase. However, the CSD and GRD and probably other solid-phase attributes, change

during that period. This first step ends with what is called the "take-off", a colloquial expression for the significant macroscopic evolution of the system. Kinetics can have a rather odd aspect, e.g., the system can remain seated on the "racemic fence" for several days before the take off. This phenomenon illustrates the stochastic aspect of spontaneous symmetry breaking. After this first period, the system shows the classical sigmoid evolution of the enantiomeric excess (e.e.) of the solid phase vs. time, which means that it possesses first-order kinetics.

The temperature versus time profile must be tuned for the achievement of the deracemization and for its productivity, together with the minimization of the chemical degradation, if there is any. The variation in solubility versus temperature is, of course, an important factor, but the cooling rate is also important for the generation of small nuclei via secondary nucleation. This phenomenon participates to the turnover of the particles of the two populations of crystals [34].

Deracemization using temperature fluctuations was demonstrated on a precursor of paclobutrazol, a molecule of interest because of its role as a plant growth inhibitor (Figure 6) [31]. In this case, the temperature fluctuations range between 20 °C and 25 °C or 30 °C and the racemization is induced by sodium hydroxide.

**Figure 6.** 1-(4-chlorophenyl)-4,4-dimethyl-2-(1H-1,2,4-triazol-1-yl)pentan-3-one, a precursor of paclobutrazol, which served as a model compound for TCID.

In addition to mechanical or thermal energy fluxes (see above), other energetic fluxes passing through the suspension lead to complete deracemization (Deracemization Induced by a Flux of Energy Crossing the Suspension (DIFECS). For instance, periodic variations in pressure [25] or pressure and temperature [35], microwaves [26], and photons for light-sensitive molecules [11] have proved to induce complete chiral symmetry breaking. This is not a limitative list. On top of this, these stimuli have agonist effects and thus can be cumulated to speed-up the macroscopic chiral symmetry breaking [36]. The common features of these processes are that they are operated somewhat close to thermodynamic equilibrium with a stochastic character regarding the final stages, R or S. They show first-order kinetics, that is, an autocatalytic global behavior. This does not mean that the predominant mechanisms are all the same. For instance, the application of ultrasound could be faster than attrition to induce complete deracemization; nevertheless, the final crystals are bigger [24]. The agonist effects of those various fluxes of energy seem more consistent with several—concomitant—possible pathways. Several mathematical models have been proposed that fit pretty well with the sigmoid shape of e.e. variation in the solid versus time [37–41]. DIFECS has been proven to be suitable for general application, provided the two following conditions are fulfilled: a conglomerate is in equilibrium with a doubly saturated solution and the chemical entities undergo rapid racemization in solution if not instantaneous—e.g., in the case of a loss of asymmetry such as $NaClO_3$ and $NaBrO_3$.

For instance, glutamic acid could be deracemized by microwave-assisted temperature cycling with a much shorter process time compared to conventional temperature cycles [26].

Kondepudi's experiment [42] is another illustration of spontaneous chiral macroscopic symmetry breaking by using crystallization in a conglomerate-forming system. Figure 7 illustrates this experiment. Typically, a racemic solution is cooled down with given kinetics in a stirred medium. If the system is able to generate a single nucleus only (the "Eve" crystal) for a sufficient period of time and if the stirring rate and stirring mode [43] are adequately tuned, numerous offspring crystals will be created

by collision with the stirrer or the wall of the reactor and thus a fast secondary nucleation originates from this "Eve" crystal. This phenomenon drops the supersaturation of the medium. At the end of the process, a homochiral population of crystals is generated, which are all descendants of the one which nucleated first. Ideally, by repeating numerous experiments, the results should be a random series of (−) and (+) populations of crystals that are statistically close to 50–50%.

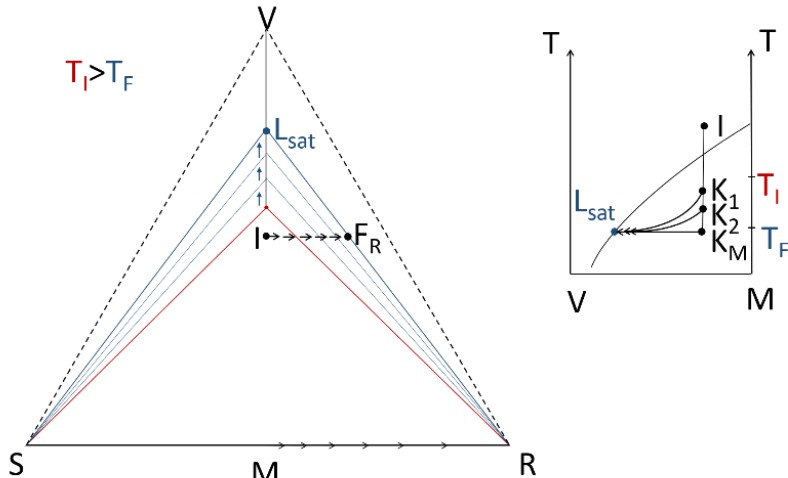

**Figure 7.** Kondepudi's experiment: in the inset, the temperature versus composition pathway of the mother liquor is represented. This is the racemic section as the liquid phase cannot deviate from enantiomeric excess (e.e.) = 0. In this experiment, it is possible that I-K1-L$_{SAT}$ and I-K2-L$_{SAT}$ pathways lead to the homochirality of the solid phase. By contrast, K$_M$ might correspond to an excessive undercooling which leads to several quasi simultaneous primary nucleation events and thus a heterochiral final population of crystals.

This process was first described by Kondepudi, who found that spontaneous symmetry breaking upon the crystallization of sodium chlorate occurs in stirred solutions, whereas static conditions give an equal amount of L and D crystals [44]. Kondepudi's experiment is also applicable to the molten state, as a supercooled melt of 1,1'-binaphthyl could crystallize with a large enantiomeric excess when vigorously stirred [45].

Preferential Enrichment (PE hereafter) also appears as a spontaneous macroscopic symmetry breaking phenomenon [46]. It was discovered and has been developed for 25 years by Professor Rui Tamura at Kyoto University. Initially confined to a series of non-racemizable organic salts (first generation), it has since been extended to other families of compounds such as amino acids and chiral pharmaceutical drugs (second generation compounds, hereafter considered non-racemizable chemical entities). It is, in a way, the opposite of preferential crystallization. Indeed, the binary system corresponds to an intermediate state between Figure 1B,C (see below for a discussion on the nature of the solid phase close to racemic composition), it is run with a considerable supersaturation and the phase that is very much enriched (>90% e.e.) is the mother liquor. By contrast, the solid phase exhibits a poor and opposite deviation (ca 4–5% e.e.). In opposition, preferential crystallization [47] is run rather close to equilibrium with a conglomerate-forming system (Figure 1A) and the very much enriched phase is the solid. The e.e. of the mother liquor is opposite to that of the solid and usually remains lower than 20% (this value is merely for the best cases). Several conditions have been pointed out in relation to the success of PE:

(i)    A large solubility difference between the racemic composition (poorly soluble) and the pure enantiomer (much more soluble) is necessary. The alpha molar ratio, $\alpha = s(\pm)/s(+) = s(\pm)/s(-)$, is thus very small (this constitutes another contrast with preferential crystallization for which $\alpha$

is usually comprised between: $\sqrt{(2)}$ and 2). Various analyses of racemic solutions in different solvents led to the surmised existence of solvated homochiral assemblies.

(ii) For a globally racemic composition, the crystal structure permits a certain degree of disorder between homochiral chains and/or planes, even if single crystals obtained from poorly supersaturated racemic solution (e.g., point $\Omega_{1.2}$ in Figure 8 representing a supersaturation $C/C_{SAT} = 1.2$) could have their structures resolved by X-ray diffraction in centrosymmetric space groups such as P-1 (the most frequent for PE) or P2$_1$/c. However, crystals obtained under high supersaturation (i.e., from a clear solution represented by point $\Omega_8$ in Figure 8) clearly reveal, by Second Harmonic Generation (SHG), homochiral domains. If this effect cannot be detected, PE experiments fail [48].

(iii) The first generation of compounds showing PE effect have all exhibited solid–solid transitions between various disordered phases. For the second generation of compounds showing PE effect in some cases, no such solid–solid transition could be detected. A solid–solid transition during PE does not appear anymore as a mandatory condition for its success.

(iv) From the first solid crystallized, which can have a high degree of stacking faults, a selective dissolution of domains containing the same enantiomer as that in excess in the solution occurs. A unique, detailed study [49] has shown that this dissolution is actually concomitant to the re-incorporation of the opposition enantiomer. It is thus the exchange of opposite enantiomers that is likely to be a concerted process. This results in a clear enrichment of the mother liquor and, simultaneously, a slight enrichment of the solid phase in the opposite enantiomer At the end of PE, the solid phase appears to be composed of Heterogeneous Nearly-Racemic Crystals (HNRC). This is different from a genuine solid solution (i.e., mixed crystals) where a random distribution of the two enantiomeric molecules is observed over the crystallographic sites. In HNRC, there are some homochiral domains of sub-micron to micron sizes.

(v) The HNRC could remain kinetically stable for months without a return to stable equilibrium if the system remains unstirred in a quiescent state.

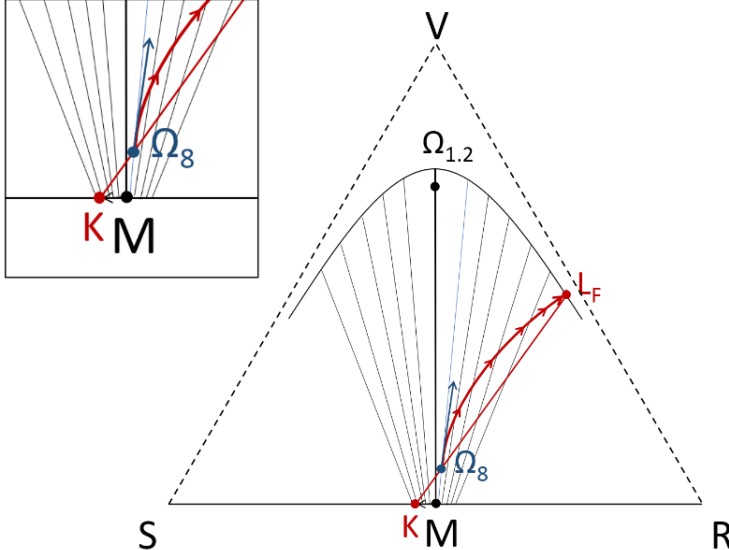

**Figure 8.** Second generation of compound presenting Preferential Enrichment (PE) effect (without polymorphism), when left very far from equilibrium, e.g., solution $\Omega_{8+}$. From the eightfold supersaturation, crystallization takes place and the solution point moves along a trajectory schematized by the red curve. The final evolution of the system is represented by the solution $L_F$ and solid K—with opposite chirality—both aligned with point $\Omega_8$, representing the overall synthetic mixture.

It is important to mention that PE has to be run at very high initial supersaturation and the system should not undergo any mechanical stress. For instance, smooth stirring introduced by the slow motion of a rocking plate is enough to derail PE and to return to normal thermodynamics without macroscopic symmetry breaking. Moreover, if the initial supersaturation is not high enough (β > four, six, eight or more (!)) PE is not observed. Thus, PE is clearly a far from equilibrium process. In Figure 8, the red tie line connecting the composition of the mother liquor and the solid at the end of the PE intersects the racemic line M–L$_{RAC}$, which is a clear violation of the thermodynamics of equilibrium. The thermodynamics of equilibrium are represented by the black tie lines and, within their context, it is impossible to have a break in symmetry, i.e., to have a solution and a solid of opposite chirality for a long period of time. When the macroscopic evolution of the system is over—this can take several days—the composition of the mother liquor (e.g., +95% e.e.) and that of the solid (e.g., ca. −4–5% e.e.) can remain unchanged for months, maybe more, in a stagnant medium.

An efficient Preferential Enrichment phenomenon could be observed for the (DL)-phenylalanine and fumaric acid co-crystal (Figure 9) [50].

**Figure 9.** Cocrystal of (DL)-phenylalanine and fumaric acid, a system that exhibits Preferential Enrichment.

There is an interesting analogy between Lamellar conglomerates [19] and fluctuations in enantiomeric composition around the racemic composition observed in PE. On the one hand, lamellar conglomerates correspond to the stacking of homochiral domains [51]. A particle, crystallized from a racemic solution, could look very much like a nice, single crystal, but could actually be constituted by the alternation of homochiral domains. At the interface of opposite domains, a 2D racemic compound is formed, but, for unclear reasons, this packing does not expand in the third direction. Quite often, the Flack parameter [52] reveals trouble in the absolute configuration assignment of the molecule and the non-linear optics of the powder show an enhanced SHG effect compared to that of a single enantiomer with the same CSD. The global composition is thus quasi-racemic, but this is not actually a single crystal. On the other hand, when operating PE with large supersaturation and under stagnant conditions (and therefore far from equilibrium thermodynamics), there are also fluctuations in the composition of solid particles. This could also be revealed by the SHG effect. Thus, the PE effect is linked to the formation of Heterogeneous Nearly-Racemic Crystals (HNRC) or, in other words, racemic compounds with local fluctuations in their enantiomeric composition. Racemic crystals that do not display the possibility of local enantiomeric fluctuation do not exhibit any PE effects.

Stirring the crystallizing suspension curbs the fluctuations in its composition and totally inhibits PE. Local deviations in the composition of the stagnant mother liquor in the vicinity of the growing surfaces are the driving forces of those phenomena. For lamellar conglomerates, when an S crystal is growing, the R enantiomer is overrepresented in the neighboring solution. The heteronucleation of R on top of the S crystal is more likely as preferential crystallization proceeds towards the end of a run (see Figure 10 and caption). The resulting epitaxy is a sort of regulatory phenomenon that diminishes the entrainment effect, i.e., drops the magnitude of the transient symmetry breaking. One study has shown that these local fluctuations could be amplified at a macroscopic scale in the mother liquor [53]. In the case of PE, the opposite effect occurs: the incorporation of the minor enantiomer in the mother liquor around the crystal and the liberation of the minor enantiomer in the solid (slightly in default in the solid) constitute an amplification of local dissymmetry (see Figure 10 and caption).

One can perceive this phenomenon as another type of ripening resulting from an initial, far from equilibrium crystallization.

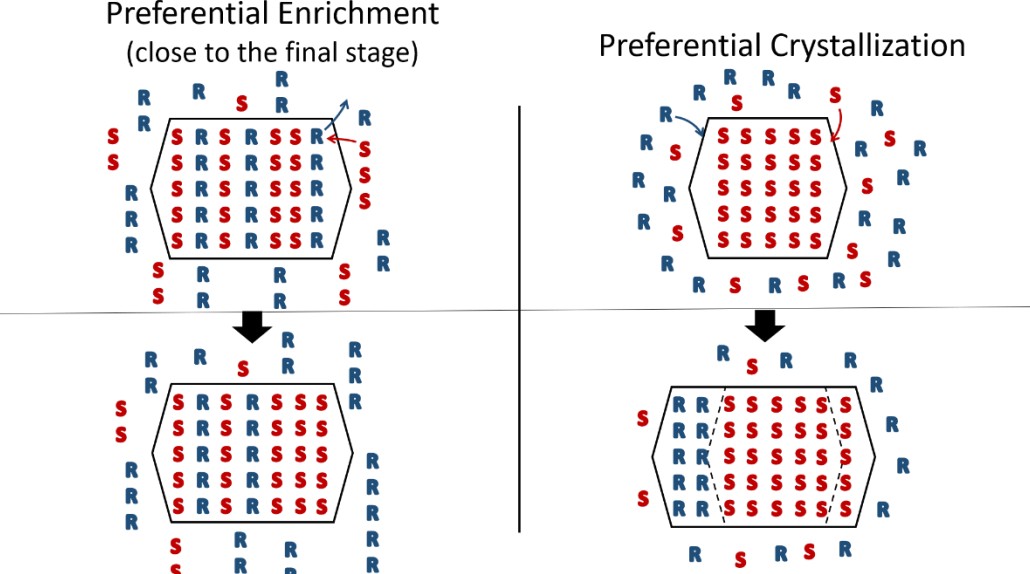

**Figure 10.** Schematic representation of the mechanisms of formation of heterochiral domains in single particles in PE and Preferential Crystallization (PC). Left: as explained in conditions for the success of PE, solvated homochiral associations are likely to exist, and are symbolized as vertical redundant letters—S or R. During the process, even when the supersaturation becomes fairly weak, the minor enantiomer in solution can substitute for the minor enantiomer in the defective solid state. Thus, there is an amplification of the e.e. in the solid and in the mother liquor. Right: focus of the crystal growth of the S enantiomer during PC, the mother liquor is depleted in the S enantiomer in the vicinity of the single crystal. As R is supersaturated, it is possible for this enantiomer to heteronucleate on top of its antipode by means of an epitaxy. In both cases PE and PC, the resulting particles are constituted by homochiral domains: sub-micron to several microns size in PE and up to hundreds of microns in PC.

One of the deep-seated reasons for those effects is the existence of an exact match between the crystal lattices of the enantiomorphous components (i.e., the Friedel and Royer conditions for the junction of the two crystal lattices are perfectly fulfilled) [54].

## 3. Control of Macroscopic Chiral Symmetry Breaking by Means of Crystallization

It is possible to orientate the symmetry breaking towards a desired enantiomer by using several robust methods. For instance, Deracemization Induced by a Flux of Energy Crossing the Suspension (DIFECS) could lead to the eutomer (the desire enantiomer) by adding a small investment prior to the beginning of the process [55]. For example, only a small percentage of e.e. (+) in the solid state is sufficient to conduct the complete deracemization by using Viedma ripening, TCID, ultrasound, microwaves, etc., towards the (+) enantiomer. This statement is valid if there is no chiral impurity in the medium that could overcompensate the initial bias introduced purposely [56], that is, without this imbalance, the system would have a stochastic behavior. This is illustrated by the Viedma ripening of sodium chloride (used as received; see Figure 11).

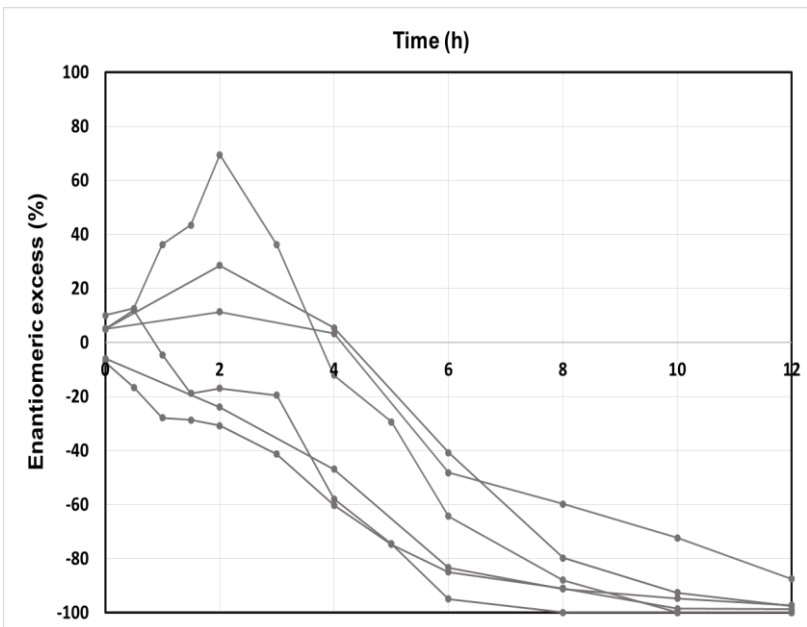

**Figure 11.** Impurity effect on the handedness of the attrition-enhanced deracemization in a non-recrystallized batch of sodium chlorate [57]. After a simple recrystallization in water, this effect disappeared.

The evolution is systematically towards the same chirality even if an initial investment in the counter enantiomer is performed. One experiment shows an evolution of up to 70% e.e. before a turn back. A simple recrystallization of the initial racemic mixture is enough to remove that effect. It is likely that a small (maybe minute) amount of a chiral impurity "pushes" the system towards the same enantiomers. This phenomenon has been observed with organic compounds [58]. Dissymmetry in crystal size distributions is also able to systematically direct the symmetry breaking towards the population of bigger crystals ("big is beautiful"). A study has shown that there is actually a balance between the initial e.e. and the dissymmetry of crystal size distributions [59]. Bigger crystals of (+) can, for instance, compensate a slight initial excess in small (−) crystals [60].

Kondepudi's experiment, seeded with very pure enantiomer prior to any primary nucleation of either enantiomer, is equivalent to preferential crystallization. A detailed analysis of the process is given elsewhere with different protocols for the inoculation of seeds and temperature profiles [47]. The symmetry of the system is purposely broken by the seeding: if the solid is the (+) enantiomer, the mother liquor evolves towards an excess of (−) in the absence of racemization. This induced symmetry breaking lasts for some minutes to some hours. The fine enantiopure inoculated crystals lead to stereoselective growth and secondary nucleation; during this period, the counter enantiomer remains in the supersaturated solution. If the system is left for too long, the second enantiomer starts to spontaneously crystallize so that the ultimate evolution of the system is a mixture of crystallized enantiomers in equilibrium with a doubly saturated racemic solution. If fast racemization takes place in the system (or in the absence of chirality in solution), the mother liquor cannot deviate from 0% e.e, as illustrated by Figure 2). In that case, the preferential crystallization receives another name: Second-Order Asymmetric Transformation (SOAT; represented in Figure 12) [61]. This elegant process could be two orders of magnitude more productive than any variant of DIFECS [62]. Supersaturation has to be kept within reasonable limits so that crystal growth and secondary nucleation remain stereoselective.

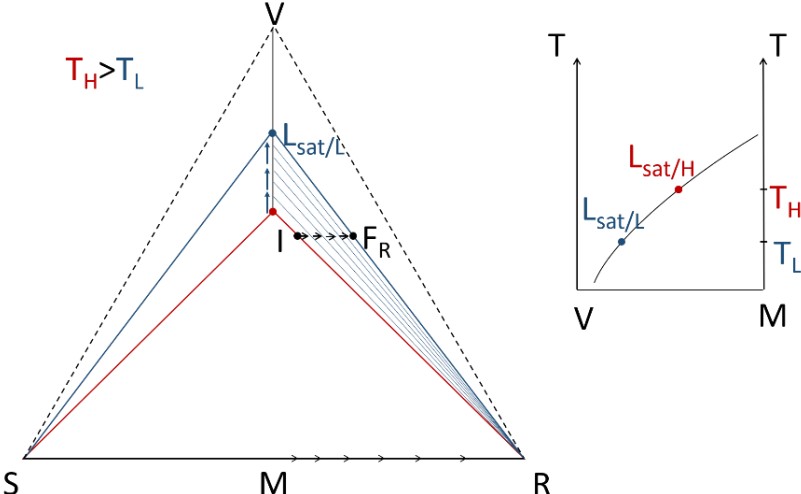

**Figure 12.** Second-Order Asymmetric Transformation (SOAT) experiment: the inset shows the temperature versus composition pathway of the mother liquor. Point I does not belong to the isopleth section represented in the inset, as the seed crystals are enantiopure.

In 1979, SOAT was used to resolve (DL)-$\alpha$-amino-$\varepsilon$-caprolactam, a lysine precursor, with the racemization being induced by potassium hydroxide in ethanol at 80 °C (Figure 13) [63]. Likewise, the enantiomers of the precursors of paclobutrazol, which served as an example for TCID, were also obtained in high enantiomeric purity with SOAT by Black et al. in 1989 [64].

**Figure 13.** (DL)-$\alpha$-amino-$\varepsilon$-caprolactam can be resolved by SOAT to produce enantiopure lysine after hydrolysis.

Lamellar epitaxies between crystals of opposite handedness could be a serious problem for Preferential Crystallization and Kondepudi's experiments [19]. The remedy is to rely on deracemization, which has been proven to achieved 100% e.e. even when the crystals of the two enantiomers can produce repeated epitaxies [65].

Kondepudi's experiments could also be controlled by the addition of specific chiral impurities. In this process, to a racemic supersaturated solution of a conglomerate-forming system, a small amount of R* additive is added to the medium. If R* has a sufficient degree of analogy with R, the nucleation and growth of R and S become significantly different and precipitation leads to an excess of S in the solid. This effect is known as the rule of reversal [66]. Of course, there is a reversibility of the rule of reversal, which means that if S* and R* also crystallize as a stable conglomerate, R could stereoselectively delay the nucleation of R*, giving way to the nucleation and growth of S*. It is also possible to induce stereoselective nucleation by using a polarized laser beam in a supersaturated solution. This process, known as Non-Photochemical Laser Induced Nucleation (NPLIN), has received some attention [67].

In the case of PE, the final state could also be controlled by an initial minor investment. As the initial step is a total dissolution of non-racemizable enantiomers (at least in the context of the crystallization), the CSD of the solid has no influence on the outcome of the process. Successive applications of PE lead to the alternation of (−), (+), (−), (+), etc., a slightly enriched solid and the opposite series for the liquid phase, strongly enriched in (+), (−), (+), (−), (+), etc. These alternations are clearly reproducible.

## 4. Conclusions and Perspectives

Crystallization offers a variety of methods for spontaneous macroscopic chiral symmetry breaking. The corresponding processes are run close to equilibrium for Deracemization Induced by a Flux of Energy Crossing the Suspension (DIFECS) (Viedma ripening, TCID and other deracemization variants where a single flux or several fluxes of energy pass through a suspension) or with a significant departure from equilibrium for the Kondepudi experiment with preferential secondary nucleation, or even very far from equilibrium in the case of Preferential Enrichment (PE). Those processes include an important amplification of local fluctuations and their mechanisms are currently the focus of many studies. Interestingly, as the system departs more from equilibrium, the mechanical stress imposed on the system has to be softened in order to observe the spontaneous chiral symmetry breaking (see Table 1). Indeed, in Viedma ripening performed close to equilibrium, all sorts of abrasions, breakages, defects induced by shear forces, etc., are beneficial to the advancement of chiral symmetry breaking. In the Kondepudi experiment, the mechanical stress must be softer to avoid primary and secondary heteronucleation in the system. In the case of Preferential Enrichment (PE), the very large departure from equilibrium has to be associated with almost stagnant conditions. Indeed, for the latter case a simple magnetic stirrer is sufficient to return the solution to normal conditions of crystallization without macroscopic chiral symmetry breaking (see Table 1 below).

**Table 1.** Spontaneous macroscopic break of symmetry by means of crystallization. In theory, the final state of the system is not predictable. There is a stochastic evolution during the first step and then amplification. In Deracemization Induced by a Flux of Energy Crossing the Suspension (DIFECS) processes, there are agonist effects between ultrasound, microwaves, attrition, TCID and damped TCID (and photons for light-sensitive molecules).

| Towards greater departure from equilibrium ↑ | | | |
|---|---|---|---|
| | Preferential Enrichment | Heterogenous Nearly-Racemic Crystals (HNRC) with possibility of alternating homochiral domains. Largest domains corresponding to the minor enantiomer | 0: Stagnant conditions to stay away from stable equilibrium | ↓ |
| ↑ | Kondepudi's Experiment | Conglomerate forming system Fast racemization in solution or non-chiral entity in solution | Moderate, collision but avoid too strong shearing effects | ↓ |
| ↑ | Deracemization Induced by a Flux of Energy Crossing the Suspension (DIFECS): -Ultrasounds -Microwaves -TCID -Viedma Ripening (close to 0) | Conglomerate forming system Fast racemization in solution or non-chiral entity in solution A lamellar conglomerate does not hinder deracemization | From soft to medium for TCID Strong with shearing effect for attrition enhanced i.e., Viedma Ripening | ↓ **Towards greater intensity of the mechanical stress** |

It is also possible to control a macroscopic chiral symmetry breaking such as Viedma ripening, TCID and its variants, or SOAT or preferential crystallization. Some of these elegant processes could be made productive enough for industrial applications. Thus, there are fundamental and applied interests in these crystallization processes associated with macroscopic symmetry breaking.

**Author Contributions:** Both authors contributed to this publication. All authors have read and agreed to the published version of the manuscript.

**Funding:** This review is part of a PhD funded by the University of Rouen Normandy.

**Conflicts of Interest:** The authors declare no conflict of interest.

## List of Symbols and Abbreviations

| | |
|---|---|
| CSD: | Crystal Size Distribution |
| DIFECS: | Deracemization Induced by a Flux of Energy Crossing the Suspension |
| e.e.: | Enantiomeric excess = (R − S)/(R + S) |
| GRD: | Growth Rate Dispersion |
| HNRC: | Heterogeneous Nearly-Racemic Crystals |
| NPLIN: | Non-Photochemical Laser Induced Nucleation |
| PC: | Preferential Crystallization |
| PE: | Preferential Enrichment |
| SOAT: | Second Order Asymmetric Transformation |
| TCID: | Temperature Cycle-Induced Deracemization |
| US: | Ultrasound |

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
