# Peer review of "Spontaneous and Controlled Macroscopic Chiral Symmetry Breaking by Means of Crystallization"

_symmetry, doi:10.3390/sym12111796_

Round 1
Reviewer 1 Report
This is a very nice contribution on chiral symmetry breaking through crystallization. Different thermodynamic and kinetic aspects are discussed. There are a lot of technical terms, which sometimes make the article less easy to read, but I believe that in this type of contribution this makes sense.
I would therefore recommend for publication.
Author Response
Thank you for these comments.
Reviewer 2 Report
The manuscript by Coquerel&Hoquante aims at summarizing various aspects of deracemization by means of crystallization. The manuscript presents a useful overview of various types of systems that can be used for this purpose. The theoretical background of the topic is well described, however, this reviewer lacks examples of specific substances, which apart from notoriously known sodium perchlorate, is not offered in the text. The authors may wish to introduce several examples of compounds that undergo spontaneous symmetry breaking for each of the described methods. Such an overview will offer the readers with handy examples and provide more specific information on applicability of the particular method on a problem the readers try to solve. Indeed, the review provides references to several topical works on spontaneous deracemization of various class of compounds; however, the lack of specific examples may discourage the readers from further reading the article.
Therefore, this reviewer suggests to improve the manuscript with specific examples for each discussed method.
Author Response
We appreciated the comments of reviewer 2 and as suggested we have added examples for every method.
Reviewer 3 Report
The subject of the review by Gérard Coquerel and Marine Hoquante “Spontaneous and Controlled Macroscopic Chiral Symmetry Breaking by Means of Crystallization”, which has a direct connection with the fundamental problem of the origin of life and with the practical needs for single enantiomeric drugs, is relevant and will find its readers. We note right away that in the present form, the review is aimed at a reader who has significant preliminary training, and such an audience is not too wide. More detailed explanations of the essence of specific concepts (such as Viedma ripening, TCID, Kondepudi experiment, Preferential Enrichment and so on) could broaden the readership. However, this question remains at the discretion of the authors and editors.
The review is a solid coverage of subtle issues, most of which have arisen relatively recently and in the study of which the group headed by Professor G. Coquerel is an undeniable authority. Therefore, a large proportion of references to their own works in the context of this review looks justified. Of course, the review can be published, but some comments can be made.
- Lines 45-49 are devoted to kryptoracemic compounds. The authors define them as structures in which “the two enantiomers are in equal amount in the unit-cell but as independent molecules”. There is a huge number of racemic compounds, in the asymmetric unit of which two or more independent molecules are presented. For a racemic compound to be a kryptoracemate (another name - false conglomerate), it must necessarily crystallize in a Sohncke space group with an even number Z’, and the independent molecules must have opposite configurations. Then the authors write “By extension if Z’ > 2 is an odd number, a two mirror–imaged scalemic compositions are obtained.” There are two inaccuracies in this proposal. Crystals of kryptoracemates exist in two mirror forms with an even Z' number. If Z' is an odd number, and the compound crystallizes in a Sohncke space group, then we are dealing with anomalous conglomerate, not kryptoracemate [Bishop, R. and M.L. Scudder, Multiple Molecules in the Asymmetric Unit (Z '> 1) and the Formation of False Conglomerate Crystal Structures. Crystal Growth & Design, 2009.9 (6): p. 2890-289; Dryzun, C. and D. Avnir, On the abundance of chiral crystals. Chemical Communications, 2012.48 (47): p. 5874-5876; Gubaidullin, A. T .; Samigullina, A. I .; Bredikhina, Z. A .; Bredikhin, A. A., Crystal structure of chiral ortho-alkyl phenyl ethers of glycerol: true racemic compound, normal, false and anomalous conglomerates within the single five-membered family. CrystEngComm, 2014.16 (29): p. 6716-6729.]
- There is an incomprehensible note on line 123.
- On line 222, the formula α = s (±) / s (-) = s (±) / s (-) should obviously look like this: α = s (±) / s (+) = s (±) / s (-)
Author Response
We thank reviewer 3 for his pertinent comments and suggested corrections.
The sentence line 123 has been rephrased.
On line 222, the formula α = s (±) / s (+) = s (±) / s (-) has been corrected (thanks for that!)
About kryptoracemic compounds and other oddities in the solid state, we agree with the terminology of reviewer 3 and we added the references suggested.
There is not enough space, you can see more details in the attachment.

Reviewer 4 Report
The proposed manuscript relies on macroscopic chiral symmetry breaking with particular emphasis on crystallization by several approaches, including Viedma ripening, Temperature Cycles Induced Deracemization or the Kondepudi experiment.
While the content of the paper is of high scientific competence and relevance, the format of the manuscript resembles much more that of a chapter in a book rather than that of a review article to be published in a peer reviewed journal. This work would be a pretty nice guide for students and early stage researchers in the field to be included in an handbook or similar. Therefore, while there is no specific scientific or technical comment on the content of this manuscript, this reviewer advices the Editor on the opportunity to publish such manuscript in its current (book chapter) form.
Author Response
We thanks reviewer 4 for his comments.
Round 2
Reviewer 4 Report
Authors revised the manuscript including some specific examples. This make the shape of the manuscript more similar to conventional reviews, although not completely fitting. However, considering the relevant scientific and didactic content of the work, that would become a guide for students, I recommend the publication of the manuscript in its current form.
Author Response
Thank you for your comments.